# Can Conformal Prediction Obtain Meaningful Safety Guarantees for ML Models?

**Emmanouil Seferis, Simon Burton, Chih-Hong Cheng**
Fraunhofer Institute for Cognitive Systems (IKS)
Hansastr. 32, Munich, Germany
`{emmanouil.seferis,simon.burton,chih-hong.cheng}@iks.fraunhofer.de`

## Abstract

Conformal Prediction (CP) has been recently proposed as a methodology to calibrate the predictions of Machine Learning (ML) models so that they can output rigorous quantification of their uncertainties. For example, one can calibrate the predictions of an ML model into prediction sets, that guarantee to cover the ground truth class with a probability larger than a specified threshold. In this paper, we study whether CP can provide strong statistical guarantees that would be required in safety-critical applications. Our evaluation on the ImageNet demonstrates that using CP over state-of-the-art models fails to deliver the required guarantees. We corroborate our results by deriving a simple connection between the CP prediction sets and top-k accuracy.

## 1 Introduction & Related Work

Deep Neural Networks (DNNs) have recently shown remarkable performance in many tasks including image and speech recognition Krizhevsky et al. (2017); Graves et al. (2013), language Brown et al. (2020), or game playing Silver et al. (2018). Despite the success, it is well known that predictions made by DNNs are heuristic and cannot offer us a notion of their uncertainties.

Recently, Conformal Prediction (CP) Angelopoulos & Bates (2021) has been proposed as a method to build statistically rigorous uncertainty sets for DNN predictions. This is done by calibrating the DNN predictions into prediction sets in such a way that the latter satisfies some statistical guarantees, for example, to contain the correct class with a probability of at least 90%. Conformal Prediction has been applied to a series of ML problems, such as classification, regression, image segmentation, and more.

In this work, we focus on the standard setting of classification prediction sets, and ask the following question: what is the efficiency of the guarantees obtained - e.g. the prediction set sizes - as we increase the required confidence levels from 90% to 99% or even 99.99%? Recall that a well-calibrated uncertainty estimate suggests that the confidence on the prediction will be the same as 1 subtracting the error probability. For reaching high safety-integrity levels (SILs) Smith & Simpson (2020) demonstrated by the need for autonomous driving, the extremely small quantity regarding the probability of dangerous failure on demand (PFoD) implies that one should achieve extremely high confidence in the prediction. This question does not seem to have been addressed in the literature, as the focus of prior works lies more on the methodology side.

In order to answer this, we evaluate several State-of-the-Art (SotA) models based on Convolutional Neural Nets (CNN) or Transformers (ViT) Dosovitskiy et al. (2020) on the ImageNet Deng et al. (2009) dataset. The results show that, aiming at high safety-integrity levels, the guaranteed prediction sets become trivial, and essentially include almost all classes of the dataset. This shows that even SotA DNNs struggle to obtain any meaningful CP guarantees at high safety-integrity levels, which may have profound implications for safety-critical domains such as autonomous driving or medical imaging.

Finally, we investigate this phenomenon theoretically, and demonstrate a simple connection between CP prediction sets and the top-k accuracy of a model, that holds regardless of the CP variant used. Essentially, we find that a DNN can obtain non-trivial prediction set sizes only if its top-k accuracy is

Table 1: Average prediction set size $|C(\hat{y})|$ for each error rate $a$

| Model / $a$ | $a = 10\%$ | $a = 5\%$ | $a = 1\%$ | $a = 0.5\%$ | $a = 0.1\%$ | $a = 0.05\%$ | $a = 0.01\%$ |
|---|---|---|---|---|---|---|---|
| ResNet-152 | 3.05 | 4.22 | 36.81 | 80.12 | 266.74 | 368.46 | 829.47 |
| WideResNet-102 | 3.33 | 4.47 | 34.62 | 76.79 | 249.19 | 384.19 | 716.52 |
| ConvNext (large) | 3.59 | 6.31 | 16.47 | 39.57 | 265.43 | 452.15 | 878.70 |
| ViT (large, 16 bit) | 3.03 | 5.14 | 43.39 | 127.24 | 714.30 | 847.68 | 991.79 |

sufficiently high. This shows that, independently of statistical wrappers such as CP, highly accurate models are necessary in order to obtain reasonable safety guarantees at high SIL situations.

## 2 Methodology & Results

In order to answer the question posed in the introduction, we evaluate various SotA models on the ImageNet dataset. We evaluate the following models: a ResNet-152 He et al. (2016), a WideResNet-102 Zagoruyko & Komodakis (2016), a ConvNext (large) Liu et al. (2022) and a ViT (large, 16 bit) Dosovitskiy et al. (2020). We split the ImageNet validation data into a calibration set of 48000 images, where we apply CP, and a test set of 2000 images, where we evaluate the resulting prediction set sizes. We use the Regularized Adaptive Prediction Sets (RAPS) CP methodology from Brown et al. (2020), which is an improvement of the Adaptive Prediction Sets (APS) method Brown et al. (2020) that guarantees much smaller prediction sets. We apply the split in calibration and test test 10 times, and report the average set sizes. The results can be seen in Table 1.

From Table 1, we observe that as the error probability $a$ approaches smaller values, the prediction set sizes become trivial, and start to include almost all possible class in the dataset (ImageNet has 1000 classes in total). It seems reasonable that there is a connection between the prediction set sizes, and the model's accuracy. Indeed, we can easily demonstrate the following:

**Proposition 1 (CP vs top-k):** Let $f_\theta : \mathbb{R}^n \to [K]$ be a model mapping inputs $\mathbf{x} \in \mathbb{R}^n$ into $K$ classes, and let $1 \leq k \leq K$ be an integer. Suppose that a CP method augmenting model predictions $\hat{y} = f_\theta(\mathbf{x})$ into prediction sets $C(\hat{y})$ of size at most $k$ achieves success rate (success by means of the real label being inside $C(\hat{y})$) of at least $1 - a$ ($a \in (0, 1)$) on a distribution $(\mathbf{x}, y) \sim D$. Then, $f_\theta$ has top-k accuracy of at least $1 - a$ on $D$.

*Proof*: Note that any CP method (e.g., APS or RAPS) will sort the output logits of $f_\theta$ from largest to smallest, and will include the corresponding class labels in a prediction set $C(\hat{y})$ using that ordering. If sets $C(\hat{y})$ have success rate at least $1 - a$, and the size $|C(\hat{y})|$ is at most $k$, that means that the true class $y$ will lie among the top-k classes with probability at least $1 - a$. But that means that the model's top-k accuracy is at least $1 - a$. This holds regardless of the specific CP methodology used.

In order to examine the implications of Proposition 1, let us go through the following example: **Example 1**: Suppose that we have an ImageNet model, and we want our prediction sets to have size at most $k = 10$ (we consider this size reasonable, since it represents no more than $1\%$ of the total $K = 1000$ classes in ImageNet). Suppose we demand that the methodology has an error probability at most $a = 0.01\%$ (which represents a SIL level of 3). Then, the model's top-k accuracy should be at least: $acc_{10} \geq 1 - a = 0.9999 = 99.99\%$. According to our knowledge, no such model exists.

Thus, although statistical wrappers such as CP can offer us sound statistical guarantees on a DNN's outputs, at higher SIL levels such guarantees will reduce to trivial, unless the underlying model has sufficient predictive accuracy.

## 3 Conclusion

In this work, we evaluated the efficiency of CP methods at high SIL levels on SotA DNNs for the ImageNet dataset. We found, both experimentally and theoretically, that the guarantees obtained become degenerate (essentially covering all possible cases) unless the models' accuracies are high enough. For higher SIL levels, the required accuracy levels would vastly exceed what has been currently achieved in the SotA. This can have serious implications on safety-critical domains such as autonomous driving or healthcare.

## URM STATEMENT

We acknowledge that all authors of this work meet the URM criteria of ICLR 2023 Tiny Papers Track.

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
