# OpenReview forum: "Can Conformal Prediction Obtain Meaningful Safety Guarantees for ML Models?"
_ICLR.cc/2023/TinyPapers — Submitted to Tiny Papers @ ICLR 2023_

### Official Review · Reviewer_pGsg · 2023-03-24

**Confidence:** 4

**Summary Of Contributions:**

Conformal prediction in practice

**Rating:**

Great Start (GS): a submission which meets some of the reviewing criteria but has room for improvement

**Strengths And Weaknesses:**

This paper discussed conformal prediction applied to neural network classifications when high coverage is required for safety reasons. The paper analysed both theoretically and empirically about this question.

It is am important question to be studied in order for conformal prediction to be meaningfully applied to DL models.

The idea is clear but I have a question regarding the theoretical result which needs clarifications, see below.

Also some typos can be found, e.g.:

"...any CP method (e.g. APS or RAPS) will **short** the output logits..": short -> sort

I suggest a careful read through again to check the typos.

**Suggested Changes:**

The way conformal prediction sorts the logits depends on the scoring function. I suggest you to clarify this, e.g., by saying the scoring function is monotonically non-decreasing in the logit values.

---

### Comment · Area_Chair_1oCU · 2023-06-05
**Ready to archive**

This work meets the threshold for archival, contains the URM statement, and is deanonymized.

---

### Meta-Review · Area_Chair_1oCU · 2023-04-06

**Recommendation:** Invite to present
**Confidence:** 3

**Metareview:**

Thank you for your submission. The reviewers have noted that this paper is studying an interesting and important problem. They have also suggested some additional details to be included in the paper that would increase the clarity of the problem and description of the theoretical results, as well as provided some minor edits. Given that it appears these changes are minor and would be fairly simple to incorporate, and the resulting paper would meet CCR requirements, I believe it meets the requirements for presenting.

**Summary:**

This paper examines the behavior of state of the art models with respect to their ability to provide safety guarentees in the presence of conformal prediction.

**Reason For Not Giving A Higher Recommendation:**

- Requires clarification on problem statement and theoretical results

**Reason For Not Giving A Lower Recommendation:**

- Requested changes are minor and once implemented the paper would meet the CCR requirements.

---

### Decision · Program_Chairs · 2023-04-07

Invite to present